# Social Aspects of Tourism Policy in the European Union. The Example of Poland and Slovakia

**Aleksander Panasiuk** * **and Ewa Wszendybył-Skulska**

Tourism and Sport Management Chair, Institute of Entrepreneurship, Jagiellonian University, 31-007 Kraków, Poland; e.wszendybyl-skulska@uj.edu.pl
* Correspondence: aleksander.panasiuk@uj.edu.pl; Tel.: +48-126-645-720

**Abstract:** Since the beginning of the 21st century, the European Union tourism policy has been increasingly focused on initiatives in the field of social tourism, which are one of the ways of achieving sustainable development in the European tourism economy. Most of the research projects that have so far been conducted in the field have focused on the benefits for its participants (subjective one: Children and youths, seniors, disabled people, people (families) with low incomes and/or unemployed, big families). However, there is a lack of research on the analysis of the place of social aspects of tourism in the general socio-economic policy of the state and, in a detailed aspect, in the sectoral policy represented by tourism policy, as well as its potential impact on the development of the national economy and meeting tourism needs of the society. The authors tried to fill this research gap in this study. The aim of the study is to differentiate the issues related to the social aspects of tourism policy from the entire socio-economic policy pursued in the European Union and selected member states (Poland and Slovakia). The article is of a theoretical–analytical–conceptual nature. Empirical research, due to the nature of its issues, was conducted with the use of qualitative research methods. The results of the conducted research showed that activities in the field of social tourism policy are conditioned by organizational solutions for the entities that undertake them, as well as economic ones, especially in the field of financing. Moreover, they made it possible to propose the concept of a model social tourism policy with an indication of its place in the European policy on the basis of the past and future EU financial perspectives.

**Keywords:** socio-economic policy; tourism economy; tourism policy; social tourism; European Union; Poland; Slovakia

**JEL Classification:** E6; H1; H53; L83; Z3

## 1. Introduction

Since the 1950s, tourism (tourism economy) has been one of the most dynamically developing industries in the global economy and national economies, determining a 10% share in the global product and over 9% share in the labor market (UN WTO) in 2019. According to further analyses, in the states selected for the research, the share of the tourism economy in the gross domestic product in 2019 was 5.9% (Poland) and 6.1% (Slovakia), respectively. Therefore, tourism is a significant element of the structure of global economy, economy of the European Union, and economies of numerous countries. The modern tourism market, in the classical sense understood as a relation between tourist supply and tourist demand, is in the process of continuous quantitative and qualitative development. A state which, through some specialized bodies, influences market processes, creating a system of regulations, which is the basis of the conducted social and economic policy, occupies a special place in the market. The role of the state on the market is to support the mechanisms of competition, influence the relations between sellers and buyers, and in this respect consumer protection in particular. State authorities should create the basis for

influencing the market structure by providing development of high-quality and innovative market offers. In countries with a social market economy, the state should also carry out activities related to consumption to enable all social groups to access basic tourism goods and services.

The EU bodies seem to recognize the role of tourism in achieving social policy goals. It is indicated by the resolutions of the European Parliament of 25 October 2011 on mobility (European Parliament 2011), or of 25 October 2015 on new challenges and concepts in promoting tourism in Europe. In the latter, the European Parliament calls for promotion and further development of products, including services that meet the specific needs of Europe's growing number of elderly and disabled people, and child- and family-friendly tourism offers. Furthermore, it is recommended for the member states to introduce a universal accessibility requirements in the tourism sector as a criterion for support as part of economic development programs (European Parliament 2015). The problematic issue of accessibility, including tourist services, is important for the EU bodies, as is evidenced by the European Accessibility Act issued in the form of the Directive (EU) 2019/882 of the European Parliament and of the Council dated on 17 April 2019 (European Parliament 2020). The European Commission, at the request of the European Parliament, also takes actions by implementing initiatives that affect social aspects in tourism, e.g., Tourism for all, whose flagship programs are Calypso and the COSME Program.

The involvement of the European Union bodies in the functioning of the tourism economy was initiated in 1985. However, the horizontal concept of tourism policy has not been developed yet, even though several attempts have been made so far (Panasiuk 2019).

The emphasis in the EU tourism policy has moved from a policy focused on supporting the economic growth of the sector to the one focused on sustainable development that is based on social cohesion and common European values (McCabe 2018, p. 37). The change has had an impact on the activities undertaken in the social field of tourism policy of individual EU countries, including Poland and Slovakia. Both countries implement systemic solutions in the field of social tourism, based to a large extent on experiences in the implementation of social programs that are the legacy of the centrally controlled economy, which can also be observed in the experiences of other countries, i.e., inter alia, Belgium, France, or Spain. When studying the issues of social tourism policy in the European Union, the context of sustainable development should be taken into consideration. It is related to taking the issue into account in the European Union programming documents.

## 2. Methodological Assumptions

The study aims to isolate issues related to the social aspects of tourism policy from the entire socio-economic policy conducted in the European Union and selected member states. The research questions were posed as follows: (1) Which of the public entities implement pro-social activities in the field of promoting access to tourism? (2) What entities from beyond the public units contribute to the social dimension of tourism policy? (3) What are the instruments used to support the entire society or selected social groups in accessing tourism? (4) Who is a direct beneficiary of the social tourism policy? The conducted study ends with an attempt to present the concept of a model social tourism policy along with an indication of its place in the European policy on the basis of the past and future EU financial perspectives. The hypothesis, according to which the scope of the tourism policy enables the state and its organs to achieve both economic and social goals, was used for the purposes of conducting the entire research, in accordance with the theoretical relationship between economic and social policy.

The article is of a theoretical–analytical–conceptual nature. The empirical research was conducted with the use of qualitative research methods. It is due to the nature of the problem, as the issues related to the scope of the tourism policy instruments used are not of quantitative character. On the other hand however, there are opportunities to present specific ways of influence of competent state authorities on tourism economy, along with determining the premises for the actions taken—goals and expected results.

The scope of the assessment of the undertaken regulatory actions is also impossible to be made in terms of value. However, in the regulatory process, the state and its authorities indicate certain funds that may be redistributed as part of the policy towards the policy recipients. On the one hand, not all public activities are of financial nature. On the other hand however, financing is scattered and does not enable obtaining full statistical data. The following research methods were also used: Critical analysis of the literature on the subject, the methods of document analysis, the methods of logical operations, heuristic methods, comparative analysis, as well as modeling as a cause and effect concept.

The main sources of the data used for the purposes of the analyzes were: Official statistics of the European Union, Poland and Slovakia, legal acts regulating the functioning of the tourism market in the EU and examined countries, programming and strategic documents concerning the functioning of the tourism market in the European Union and the selected countries, information from the publications of the institutions that regulate the tourism market and other cooperating public entities and institutions that organize and finance social tourism in Poland and Slovakia, including strategies and programs.

Poland and Slovakia were selected to present detailed solutions related to the aspects of social policy in the European Union countries. On the one hand, it was made from the point of view of access to empirical material, but above all due to the fact that the criteria of comparative analysis can be applied directly to both countries, which have been members of the European Union since 2004. The political and social systems of Poland and Slovakia have undergone analogous transformation processes since 1989. Many elements of social policy in post-socialist countries use pre-defined, usually well-established instruments of its implementation. The analysis will cover the years after both countries joined the structures of the European Union, with particular emphasis on the years 2014–2019. The vast majority of tourism policy instruments in terms of achieving social goals implemented independently in both Poland and Slovakia have been transferred or modified from the realities of the socialist economy.

In the course of the research, the following research steps concerning the social tourism policy were carried out: (1) Formulation of the research problem, (2) development of the theoretical and empirical research concepts, (3) analysis of theoretical literature sources, (4) analysis of legal acts, programming, and strategic documents of the European Union and countries that apply solutions in the discussed subject (e.g., Hungary, Portugal), Poland and Slovakia in particular, (5) analysis of the tourism policy system of the European Union, Poland, and Slovakia, (6) synthetic analysis of the state of development of Polish and Slovakian tourism economies and level of consumption, (7) an attempt to identify the applied instruments of social tourism policy implemented by the public entities and institutions cooperating with one another in the analyzed countries, (8) an attempt to generalize the analyzes by constructing a structural model of social tourism policy, and (9) formulating a conclusion concerning, inter alia, European social tourism policy in the context of funds available for tourism in the next financial perspectives.

## 3. Theoretical Background. Review of Literature

### 3.1. Socio-Economic Policy

Functioning of the modern market is based on the economic order, resulting from the adaptation of various institutions, mechanisms, and behaviors of economic entities to the changing economic factors, and external circumstances toward the existing economic system. Such an order may also be a consequence of the regulatory actions of the country, consisting in preventing negative phenomena resulting from the imperfect structure of the market (Surdej 2006). In the broadest sense, the concepts of regulation and regulating are defined as forms of social control of economic activity in an institutionalized or non-institutionalized form that is carried out informally or formally with the use of specific instruments (Boyer 1986). Market processes constitute an important area of the interest of the country, which is manifested in the creation of a regulatory system. These days, the most important ways by which public authorities, at the national and supranational level,

intervene in the economy are competition law and sector regulation. Both of these areas of interference derive their genesis and justification for their existence from the common belief that the economy cannot meet human needs in an effective way without appropriate legal norms that prevent the emergence of or correct specific imperfections of the market mechanism. The modern economic system can be called a "state of regulation" in which the state, giving up its ownership of production factors, does not give up control over the economy (Majone 1994; Goczek 2012). Regulation is implemented through the conducted socio-economic policy, provided that the necessary level of market freedom is ensured. Market regulations can be introduced in order to reduce market failure (interventionism) and to promote and protect competition (liberalism) (Baldwin and Cave 1999; Black 2001).

The set of actions undertaken by the state in order to implement the economic order and goals related to it is referred to as socio-economic policy. It should consist in creating and protecting conditions for the functioning of the mechanism of automatic adjustment of the volume of supply to demand through prices. However, it requires the protection of property rights and settlement of contractual disputes, as well as the existence of many independent producers, well-informed and protected consumers, free pricing, and freedom to enter and exit the market. Creating such conditions requires many regulations conducted by state institutions, which are of a nature as follows (Firlit-Fesnak and Szylko-Skoczny 2007; Horodecka 2008):

- economic—related to the functioning of market entities, and above all related to the impact on the economy, its dynamics, structure, and condition of economic relations with foreign countries,
- social—covering the impact on social needs related to the accessibility to basic goods and services, as well as social security and labor, health, education and upbringing, housing and culture, and system constituting a mechanism for securing public order and safety.

Economic policy affects the functioning of the economic system of the country, which is a complex of organizations, households, and units that operate according to certain rules, incentives, orders and prohibitions in the field of production, distribution, exchange, and consumption of material goods and services (Kowalik 2000). Contemporary social policy is aimed at securing both elementary needs and those that are satisfied by established consumption patterns in the conditions of the achieved civilization level (Baldock et al. 2012). Such needs include tourism needs. Through the concept of socio-economic policy, particular attention is paid to the need to combine social and economic goals and conditions of state interference in economic affairs. As part of the implemented economic policy, the state has an ability to shape the conditions, including, inter alia, legal norms leading to the sustainable development of the entire national economy or its individual sectors (González 2005; Staiger and Wolak 1994).

*3.2. Tourism Policy*

Influence of the state within its policy applies to many areas and social aspects of the economic process, which entails the need for precise identification of the problems that occur in these areas, and then the need for proper specification of the methods and means of interference (Joppe 2018). Therefore, as part of the entire socio-economic policy, various spheres of impact and subsystems develop, e.g., macro- and microeconomic policy, international policy, regional policy, and policy in individual economic sectors (Winiarski 2006). One of the examples of sectoral policy is tourism policy, which makes it possible for the country to directly influence tourism economy (Scott 2011). It also manifests itself in other profiled sectoral policies, particularly in international and regional, industrial, employment, environmental, cultural, commercial, and communication policies.

State interference in the tourism economy is aimed at: Demand, supply, shaping the position of tourism economy in the national economy and its relations with other elements of the economy structure. Tourism policy as a sectoral policy is an activity consisting in defining economic, political, social, and cultural goals related to the development of

tourism, obtaining comprehensive positive effects resulting from the existence of demand and supply, striving to satisfy social needs in the field of tourism and specifying the necessary measures (Wodejko 1998). In a fuller aspect, tourism policy should be defined as the activity of the state and its organs consisting in defining economic and social goals related to tourism, as well as the selection of appropriate instruments. These are necessary for their implementation, leading to shaping the structure of the tourism market, both in terms of tourism supply and tourism demand, and in the supply-demand relationship (Panasiuk 2019). Activities undertaken by the state authorities in the field of tourism policy of a social nature should lead to, inter alia: Meeting tourism needs of the society and shaping an optimal size and structure of tourism traffic (Kurek 2008). Therefore, the basic function of the state is to create economic and non-economic goals, as well as to select the means (instruments) necessary for their implementation. The structure of tourism policy consisting of: State authorities implementing the policy (policy entities), tourism entrepreneurs and offers of such, as well as tourism consumers (policy recipients), model solutions for the role of the state in the economy (policy methods), and policy instruments (Panasiuk 2011) should be defined.

The basis for the structure of the socio-economic policy, including tourism policy as well, is the accepted model (method) of policy. Then, the implementation of the policy is made through appropriate instruments (tools) subordinated to the applied model of policy, which are the means of achieving goals and tasks. Contemporary tourism policy, mainly the European one in terms of declarations, accepts the formula of a mixed model of moving away from far-reaching interventionism towards liberalism. Such a model is called deregulation. In other words, it means limiting the state's functions in the economy (McGuigan et al. 2008). A consequence of it may be adoption of the concept of economy management solely on market principles (Bramwell and Lane 2010).

The instruments of tourism policy, used by competent entities and subordinated to the objectives of its implementation, constitute a concrete form of acceptance of a specific policy model by the state. Tourism policy instruments can be classified according to two criteria. On the grounds of the multi-level system of regulations, the groups of tourism policy instruments can be distinguished as follows (Panasiuk 2017):

- General instruments of the state policy—resulting from the entire socio-economic policy of the state (e.g., stimulation of the service sector, activities in regional development matters, shaping the cross-border cooperation),
- detailed instruments of tourism market—related to a comprehensive impact on the tourism market, resulting from tourism legislation,
- specific (specialized) instruments, i.e., those related to the regulation of individual tourism submarkets (e.g., administrative regulations on accommodation, access regulations for entities of the travel agency market, rules for foreign travel).

In terms of types, the instruments of tourism policy include: Economic, legal, administrative, organizational, informational, and moral ones (Bosiacki and Panasiuk 2017). The selection of appropriate policy instruments allows to achieve the assumed economic and social goals.

*3.3. The Place of Tourism in the Socio-Economic Policy of the European Union in the Member States*

Tourism economy as the area of interest of socio-economic policy constitutes the basis for making some decisions of a market nature by competent entities of tourism policy. These activities are focused mainly on the supply side of the market. An important area of the state bodies is also influencing tourism demand, which results from the following premises of:

a. General socio-economic development, and thus treating tourism needs as those reported universally and in the case of certain forms of tourism of a basic nature,
b. increasing diversification of purchasing abilities of households for non-standard products,

c.    better use of tourist infrastructure, and
d.    shaping macroeconomic effect (impact on the labor market, fiscal policy) (Panasiuk 2012).

The main area of the state's impact on tourist demand should be shaping tourist activity of the society. The process of influencing the inbound tourist traffic has the features of a pro-demand policy. However, it concerns the indirect aspects only, as it concentrates on supporting the supply system, the development of tourist and para-tourism infrastructure, creating a tourist offer and its promotion in particular. These activities are to create a demand effect among tourists arriving in the area, i.e., those who may be motivated by the policy that supports demand from the place of issue (Panasiuk et al. 2016).

As part of the accepted social policy, the state may create conditions for the universal use of tourism. Thus, it may simplify access to the selected offers of the tourism market in the following two approaches (Tureac and Turtureanu 2010; Minnaert et al. 2006):

(a)    subjective one:

- Children and youths,
- seniors,
- disabled people,
- employees working in hazardous conditions,
- inhabitants of ecologically endangered areas,
- people (families) with low incomes and/or unemployed,
- big families,

(b)    subject one, e.g.,

- social tourism,
- health tourism (including wellness and spa tourism),
- family tourism,
- educational tourism,
- cultural tourism,
- active tourism,
- ecotourism.

Tourism economy is not widely regulated by the European Union bodies. In terms of tourism policy, the member states have almost full autonomy. On this basis, it should be stated that there is no horizontal tourism policy in the sense of other policies in the European Union (e.g., agricultural policy). The choice of a detailed model and, subsequently, instruments of tourism policy in the EU member states is the responsibility of every domestic authority. The concept of tourism policy at the level of individual countries is consistent with the general system of socio-economic policy. The most important manifestation of the common tourism policy of the European Union is regulations covering the tour operator market, including consumer protection issues that result from the general principles of the functioning of the European Union and, above all, from the Directive (EU) 2015/2302 of the European Parliament and of the Council of the European Union of 25 November 2015 on tourist events and related tourism services (Directive 2015), leading to the absolute harmonization of the European tourism market as regards the activities of travel agencies and entities providing related tourism services. The directive extended the scope of regulation of the market in relation to the existing solutions and strengthened the position of the client (Panasiuk 2018).

As it has already been mentioned in the Introduction, the issues of social tourism policy should be based on the concept of sustainable development. Such an approach is closely related to documents concerning tourism policy in the European Union. The essence of sustainable development comes down to shaping a development that should enable us to satisfy today's needs without limiting the chances for future generations (Adamczyk and Nitkiewicz 2008). Therefore, the basis for sustainable development is a development that meets the needs of modern times, including the needs of future generations. The concept of sustainable tourism was created as a result of research into the relationship between

tourism, environment, and development (Panasiuk 2020c). It also means providing tourists with a better experience and tourism companies with more opportunities to develop themselves (Pender and Sharpley 2008; Nijkamp and Verdonkschot 2000). The objectives of social tourism policy should be consistent with implementation of tasks resulting from the concept of sustainable tourism development (Ritchie and Crouch 2003), as tourism policy is based on taking into account the combined economic, social, and environmental aspects. It also creates competitiveness (Estol et al. 2018; Estol and Font 2016; Paunović et al. 2020).

### 3.4. Manifestations of Social Aspects in Tourism Policy

However, there is still no common EU tourism policy for social cohesion. It results from the provisions of the Lisbon Treaty of 2009, which, despite the fact that it excludes the harmonization of laws and regulations regarding tourism in the member states, allows the EU bodies to support, coordinate, and supplement them in the area of tourism policies (Juul 2015).

Over the years, the European Union has changed its orientation towards tourism, particularly the one which takes social aspects into account. A sign of the European Commission's interest in the social aspect of tourism was the comparison of the so-called social tourism in individual EU countries in the 1990s (Unite' Tourisme—Commission des Communaute's Europe´Ennes—D.G. XXIII 1994). Moreover, in 2002, the European Economic and Social Committee on Social Tourism in Europe explained in its opinion (European Economic and Social Committee 2006, art. 2.2.2) the understanding of social tourism, indicating that it must meet three conditions:

1. Real-life circumstances are such that it is totally or partially impossible to fully exercise the right to tourism. It may be due to economic conditions, physical or mental disability, personal or family isolation, reduced mobility, geographical difficulties, and a wide variety of causes which ultimately constitute a real obstacle.
2. Someone—whether a public or private institution, a company, a trade union, or simply an organized group of people—decides to take action to overcome or reduce the obstacle which prevents a person from exercising their right to tourism,
3. The action is effective and actually helps a group of people to participate in tourism in a manner which respects the values of sustainability, accessibility, and solidarity.

However, the need for sustainable tourism growth within the European Union, which would balance job creation, was noticed by the European Commission only in 2006 with the publication *A renewed EU Tourism Policy: Towards a stronger partnership for European Tourism* (European Commission 2006). The renewed Tourism Policy of Europe was introduced in 2007 with the publication of *Action for more sustainable European tourism* report by the Tourism Sustainability Group (TSG), which indicated three goals, including one assuming equality and social cohesion (Denman and Mihalič 2007, p. 3).

However, this goal did not refer directly to the social aspect of tourism policy. Instead, it referred only to the improvement of socio-cultural life of host communities and their involvement in tourism development in their region and/or to provide a safe or fulfilling experience for all visitors without discrimination (Diekmann and McCabe 2011, p. 422). Only the conferences organized by the Tourism Unit in cooperation with ISTO (International Social Tourism Organization) clearly emphasized the necessity to take social aspects into account in the EU tourism policy. The first one (2006) referred to the issue of "Tourism for all" which is important for the EU cohesion policy. Then, the current situation and practices in the field of social tourism implemented in the EU countries were discussed. The participants of the conference recognized that the concept of "Tourism for all" can be an excellent tool to solve the biggest problems of the European tourism sector, such as congestion and seasonality. Another conference took place in 2007 as part of the celebration of the "European Year of Equal Opportunities for All". Its main goal was "to identify whether there exists the possibility of extending collaboration on social tourism in different Member States that are currently less active than others in this field" (European Commission 2008). The 2008 conference concerned social tourism in the EU, mainly tourism of youth

and seniors. During the conference, the procedures lead to the establishment of a pilot tourism project for seniors. Its purpose was to help solve problems related to seasonality in the tourism sector which, in 2009, led to implementation of the EU Calypso flagship project "Tourism for all". It has been continued under the broader COSME program (Framework Program for the Competitiveness of Enterprises and Small and Medium-sized Enterprises in 2014–2020).

The social aspects of tourism policy were and still are also important for the European Union Parliament. It is reflected in the resolutions it issued, i.e., inter alia, the one of 27 September 2011 on Europe as the most popular tourist destination in the world—a new political framework for the European tourism sector, in which the European Commission is called on to promote a gradual departure from the seasonal nature of the tourism offer and to continue making efforts to increase accessibility of the tourism offer to all EU citizens (2010/2206(INI)). Another resolution of 29 October 2015 on new challenges and concepts for promoting tourism in Europe calls on the member states to place particular emphasis on the use of new technologies while formulating concepts of tourism for seniors and people with reduced mobility, and recommends introducing universal accessibility as a criterion for granting support a part of economic development programs (2014/2241(INI)). The European Union bodies and institutions increasingly recognize the necessity to take into account the social aspects in the EU tourism policy. However, specific and consistent actions at the EU level are limited in this respect.

Starting from the EU financial perspective for 2007–2013, at the level of the European Commission and in a few member states, some actions have been taken in order to reach certain social groups with the tourist offer, allowing to some extent to eliminate social inequalities in access to tourism. The main assumption of the initiative is to support financing out-of-season tourist trips for people belonging to four social groups that have problems with self-financing of tourist needs, or those who are afraid of challenges related to the organization of a tourist trip. The target of the program was consumer segments of the European Union: Between 18 and 30 years of age, families with financial difficulties, people with disabilities and, above all, seniors. The program covered the regions of France, Portugal, Italy, and Spain. Forty-five thousand Europeans benefited from it in the first edition of the Calypso Project only (2009/10). The highest amounts of subsidies, i.e., EUR 150, were received by the inhabitants of Czechia, Poland, Slovakia, and Hungary (eCalypso 2016). "In 2009-11, the EU allocated € 1–1.5 million per year for Calypso projects and for 2012, € 450,000. For its senior tourism initiative, the EU allocated € 1 million in 2013" (Juul 2015, p. 17).

Another program of a similar scope is COS-TFLOWS-2014-3-15 (Facilitating EU transnational tourism flows for seniors and young people in the low and medium seasons). The project was elaborated by the EASME agency (Executive Agency for Small and Medium-sized Enterprises) operating within the European Commission, and today it is part of COSME (Framework Program for the Competitiveness of Enterprises and Small and Medium-sized Enterprises in 2014–2020). One of the assumptions of the project is tourism support, including strengthening competitiveness of the European tourism sector by extending the tourist season by increasing the mobility of seniors (55+) and young people (15–29 years of age). "In 2014, the Commission extended the scope of the former senior tourism initiative to cover young people too and allocated a budget of € 1.8 million" (Juul 2015, p. 17).

As part of the initiative, the European Commission has funded a great number of projects to facilitate holiday travel for groups at a disadvantage to increase off-season tourist traffic to help local economies fight seasonality by creating new jobs and business opportunities. These included (European Commission 2016):

1. European Senior Travelers (ETS): Promoting senior exchanges between Portugal, Spain, and Poland.
2. Social Tourism European Exchanges Platform (STEEP): Facilitations for international tourism, out of season in particular.

3. Holiday 4all: Strengthening and promotion of transnational cooperation in the Danube macroregion in the development of social tourism.
4. OFF2013: Facilitation of an off-season international exchange of seniors and families at a disadvantage in Hungary and Poland.

Social tourism was initially based on social assistance aimed at increasing tourist activity of elderly, disabled, and poverty-stricken people. However, its contribution to the economic development of regions has been appreciated over time. Social tourism contributes to social well-being, increases self-confidence and perspectives for the future, and relieves isolation, stress, and loneliness (Morgan et al. 2015). However, according to Cisneros-Martínez et al. (2018), it allows the direct implementation of the basic social goal of improving the quality of life of the elderly, the disabled, and the poor, by enabling them to discover new places and perform specific tourist activities, and therefore enriching free time. Social tourism offers many solutions to make it possible to spread the demand more evenly (Cisneros-Martínez et al. 2018), thus contributing to the improvement of tourists' life quality (Berbeka 2014; Estrada-González 2017) and economic sustainability of resorts and regions. It helps tourist regions to maintain economic activity and employment during the off-season, favoring both medium and long-term profitability. Moreover, it also contributes to the improvement of life quality of local communities. A more dynamic local economy may also lead to development of additional markets also out of season (Cisneros-Martínez et al. 2018; Ministerstwo Sportu i Turystyki 2010).

From an economic point of view, social tourism plays an important role in creating jobs (ISTO after Diekmann and McCabe 2011, p. 420) and mitigating negative effects of seasonality (European Commission 2010; Cisneros-Martínez et al. 2018; Eusébio et al. 2016). The effects of the Spanish IMSERSO program or the Portuguese INATEL program, as well as the European one—Calypso provide the evidence. The Spanish program, according to the assessments of PricewaterhouseCoopers (2012) and Instituto de Mayores y Servicios Sociales (2014), as financially sustainable in terms of the generated savings (on unemployment benefits and other benefits) and income (VAT, income tax, etc.) allowed for the reimbursement of the incurred expenditure. Additionally, the program allowed them to generate and/or maintain 119,000 jobs (including 16,000 direct and 103,000 indirect jobs) (IMSERSO 2016 after Cisneros-Martínez et al. 2018). Similar benefits, i.e., positive results in profitability, employment, the level of hotel use in the low season, and increased customer loyalty, were caused by the implementation of the Portuguese INATEL program, as it was indicated by Eusébio et al. (2013, 2016) in their research. They claimed that the economic contribution to the destinations exceeded the cost of the program based on the input–output model of the Portuguese economy, and that social tourism can help to diversify and restructure the target economy, in addition to creating jobs, generating income, and contributing to the development of destination. The issue of economic sustainability is significant, according to Agarwal and Brunt (2006), as the decline in the number of visitors to coastal destinations leads to a decline in living standards, as well as a decline in investment and infrastructure, and the restructuring of resorts towards a low-wage economy and social exclusion.

Furthermore, the effects of the Calypso program strongly justify the need to develop social tourism as the one that provides economic benefits to destinations and countries (income from higher tourism spending, increased employment, permanent employment, increased tax revenues, etc.) (eCalypso 2016).

Social tourism development also contributes to the elimination of problems related to the accessibility of tourist objects to the needs of more demanding customers (Diekmann et al. 2009, p. 29), as it requires adapting the service offer to the needs of the elderly as well as to the disabled and families with children.

*3.5. Combining Economic and Social Goals in Tourism Policy*

To sum up, it should be noted that the structure of activities undertaken by the state as part of the socio-economic policy implies the need to combine economic and social goals

(Bouchard 2009). The state, under the policy, is able to create conditions, including, inter alia, legal norms leading to sustainable development of the entire economy of the state or its individual sectors and industries, as well as social goals.

Assuming that the main task of the contemporary state policy is to implement the idea of responsibility for the welfare of its citizens, social and economic policy become integral, which should also be expressed in tourism policy, under which the objectives related to influencing the size and structure of tourism demand can be achieved through activities aimed at tourism supply, i.e., tourism enterprises and tourism destinations.

Social tourism (Minnaert et al. 2009), understood as state activities aimed at supporting tourism activity of certain social groups and in the scope of certain forms of tourism, is a direct effect of combining economic and social issues in the pursued social policy (McCabe 2009). In other words, social tourism is a support activity based on a subsidy from the public sector aimed at including individual groups that have not participated in tourism so far (Włodarczyk 2010). The state's activity in the field of social tourism should lead to an increase in the net tourism activity indicator, which, in Poland, is estimated at around 60%, which is the percentage of the inhabitants participating in at least one tourist trip a year, regardless of the number of nights. Thus, it should be stated that about 40% of Poles are covered by the so-called tourist exclusion (Berbeka 2016).

## 4. Tourism Policy Review in Terms of Social Goals

### 4.1. Poland

Poland is said to be a country with a high level of tourist attractiveness. In 2019, over 20 million foreigners visited the country, which gives it the 19th place in the global ranking (UN WTO). Outbound tourism revenues are estimated at $16.8 billion. The number of domestic trips amounts to approximately 48 million, and tourist expenses are estimated at approximately 31 billion PLN (approx. $8 billion). The number of overnight stays is 18.7 million (Eurostat 2020b). The total number of beds in Poland is approximately 825.5 thousand. Between 2005 and 2019, the above-mentioned values were characterized by a constant and high growth dynamics. It should be emphasized that tourism balance, i.e., surplus of revenues from incoming foreign tourism in relation to Poles' expenditure on outbound tourism, is favorable.

Table 1 shows the average disposable income of Poles, which, despite the systematic increase, is below 50% of the income in the European Union and is also lower than the income of Slovaks.

**Table 1.** The average disposable income of Poles in 2014–2019.

|  | 2014 | 2015 | 2016 | 2017 | 2018 | 2019 |
|---|---|---|---|---|---|---|
| GDP per capita in PPS [%] | 41.4 | 42.2 | 42.7 | 43.6 | 45.1 | 46.4 |
| Average disposable income (Euro) | 10.51 | 10.92 | 11.26 | 11.82 | 12.46 | 12.98 |

Source: Eurostat (2020a).

Poles' tourist consumption is dominated by short-term (weekend) trips around and across the country. In 2018, it was about 28.9 million trips. It increased by 4.5% in comparison to 2017. The number of long-term (holiday) trips increased to 18.8 million, which constituted an increase of 14.5%. On the other hand, the number of foreign trips was 12.7 million and it increased by 6.6%. In 2018, there was an increase in total expenditure in the tourism economy by 6.8% and a stable share of tourism economy in the creation of GDP at the level of 5.9% (MSiT).

Gross tourist activity of Poles in domestic traffic amounts to approx. 1.24, which is about twice as high as that of Slovaks. The indicator means that each Pole goes on a tourist trip in the country more than once a year, or there are about 5 domestic trips per year for four Poles. About 20 million Poles travel for private purposes. Table 2 presents the net tourist activity rate.

**Table 2.** Poles' tourist activity in 2014–2018 compared to the activity of the citizens of 27 European Union member states in 2014–2018 (%).

|  | **2014** | **2015** | **2016** | **2017** | **2018** |
|---|---|---|---|---|---|
| EU 27 | 59.2 | 61.0 | 62.3 | 62.4 | 64.4 |
| Poland | 53.1 | 54.1 | 56.8 | 59.2 | 61.9 |

Source: Eurostat (2020b).

In 2018, 61.9% of Poles went on a trip at least once a year. It is about 8.8 percentage points more than in 2014, but 2.5 percentage points less than the average in the 27 EU member states.

The formal and legal framework of the functioning of the tourism market in Poland is determined by the legislative bodies. The legal system in Poland is adjusted to the rules in force in the European Union, particularly to the Directive of the Parliament and Council 2015/2302 of 25 November 2015 on package travel and related travel services [Directive]. Figure 1 shows the structure of entities that form the national tourism policy system.

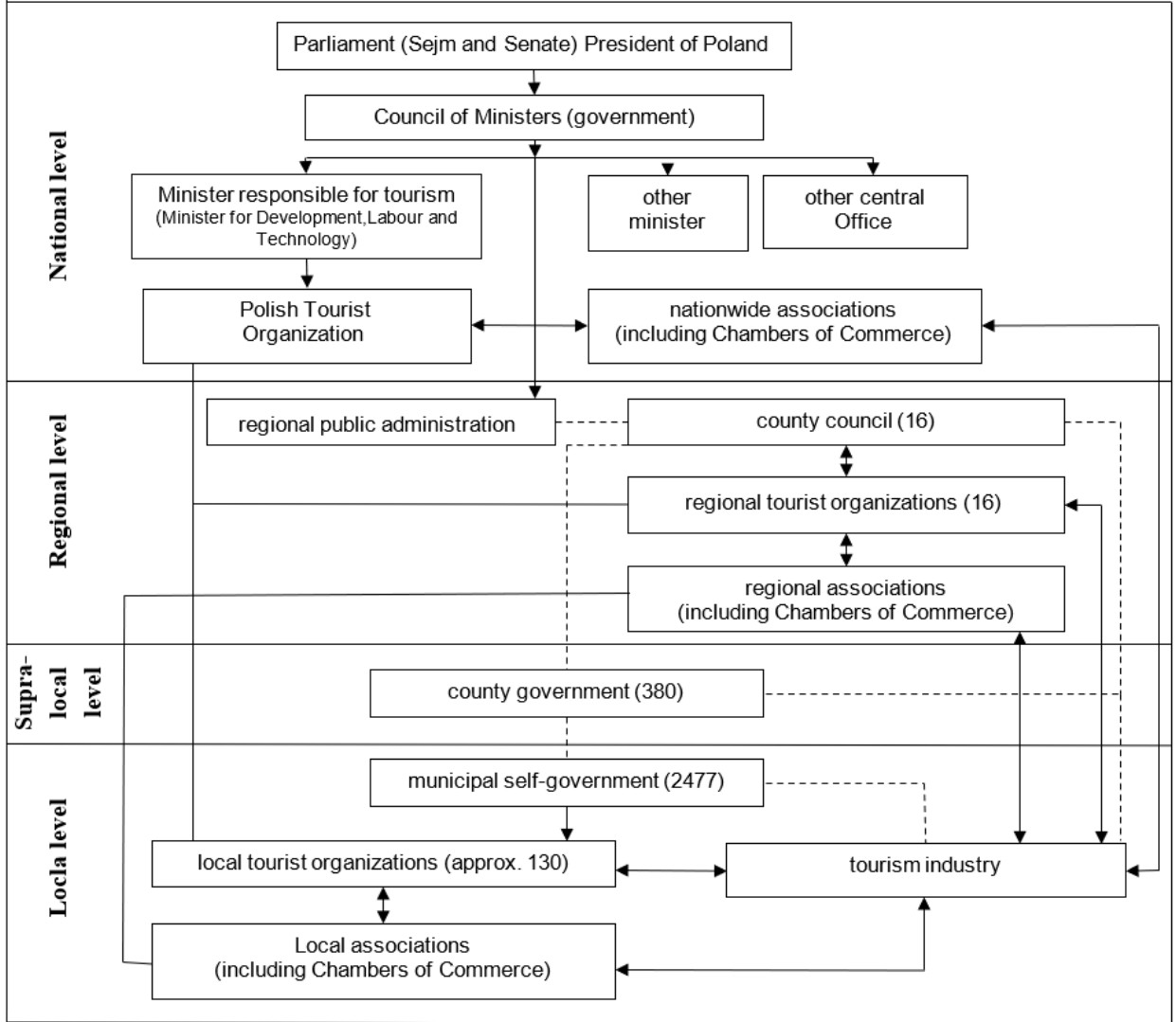

**Figure 1.** A structural system of tourism policy entities in Poland. Source: Author's own elaboration.

The entity responsible at the central level for tourism policy is the minister responsible for tourism—until November 2019, it was the Minister of Sport and Tourism, currently the Minister of Development, Labor, and Technology. The basic competences of the minister

include: Implementation of tasks related to creating development and directions of tourism promotion; development, implementation, and monitoring of tourism programs; carrying out works related to state's tourism development; programming and conducting matters related to tourists' safety; conducting the issues related to the creation of tourist market regulation mechanisms, particularly in the field of entrepreneurship development, quality improvement, and assessment of the functioning of the tourism economy, supervision over the national tourist organization, cooperation with the units that educate staff for tourism, carrying out activities in the field of tourism support from the EU structural funds. Practically, none of the tasks of the office refers to the issues of supporting the demand, except for:

- Supporting activities for the leisure of children and youth, which is also the task of the minister responsible for physical education and sport;
- consumer rights protection, which is a general instrument that regulates relations on the tourist service market.

Other central offices, including in particular: The health resort in the field of spa tourism, by co-financing stays on health stays in spas and the national education department supporting tourism for children and youth, especially school tourism, along with the so-called "Schools in Nature" for children and young people who live in areas of the country in health hazard conditions play a complementary role in government activities related to the policy of supporting tourist demand.

The Polish Tourist Organization has limited competences as an entity in the regulatory sphere. Its basic functions relate to tasks in the real sphere as an entity that implements tasks for the tourist destination of Poland, concerning, apart from marketing activities undertaken on both foreign and domestic markets, initiating, giving opinions, and supporting plans for development and modernization of tourism infrastructure, as well as cooperation with regional and local tourist organizations.

Local government units of all three levels perform both regulatory and real functions on the tourism market. Local government is obliged to implement tasks for the benefit of tourism for children and youths. The task is carried out at the level of educational institutions by the commune and powiat self-governments, as founding institutions for primary and secondary education, respectively, and comes down to the organization by the above-mentioned institutions and possibly subsidizing the Schools in Nature. The activities are also carried out by scouts and youth organizations as part of school tourism. The Polish Society of Youth Hostels plays a special role for the promotion of school and youth tourism. The organization exploits accommodation facilities and offer services at affordable prices for both tourist groups and for the possibility of family tourism.

Activities in the field of social policy, implemented at the level of enterprises and institutions, are forms of co-financing as part of corporate social benefits funds that apply to organization of leisure in company facilities for employees and their families, and organization of recreation for employees' children, subsidies for holidays as part of school tourism. The latter is not very effective as the money received by employees is not always spent on tourism. Social assistance for employees is also offered by trade union organizations.

The above information proves that the activities of state authorities in the area of direct impact on tourist activity in Poland are significantly limited. It is also confirmed by the records of the strategic documents in force in the field of tourism economy in the European Union financing perspectives for 2007–2013 and 2014–2020.

For many years, there has been a discussion in Poland on the introduction of common assistance in the field of domestic tourism and, at the same time, supporting the tourism economy through a tourist voucher. Despite many years of discussion, until 2020 it was not possible to develop a common position of entities from the tourism policy and the tourism industry and to secure public funds for this purpose. Such activities allow them to support the demand directly, and above all to support tourism economy entities, where such a voucher could be redeemed. The tourist benefit financed by the voucher has a significant advantage, as it allows voucher holders to choose the service that suits them best (Kopeć

2018).

It was not until the deepening crisis of the tourism industry caused by the COVID-19 pandemic that the Polish government decided to launch the instrument, under the name of the Polish Tourist Voucher. The instrument is assumed to provide financial support to Polish families, through means of a holiday subsidy for every child before 18 years of age in the amount of 500 PLN (i.e., approximately 110 EURO) and to establish a double amount for disabled children. The voucher covers the costs of hotel services or tourist events by a tourist entrepreneur or a public benefit organization in Poland. The tourist voucher may be redeemed between 1 August 2020 and 31 March 2022 in several thousand entities of the tourism economy. The subsidy for summer or winter holidays was given to around 2.4 million families with over 6 million children (Tourist Voucher 2020). The instrument is characterized by all social policy activities. However, in practice its main goal is to save the condition of functioning of the tourism industry during the epidemic.

*4.2. Slovakia*

Slovakia is becoming an increasingly attractive country for tourists. The statistics have proven a systematic increase in the number of tourists since 2014. In 2018, the tourist traffic rate exceeded 5.5 million tourists, 60% of whom are domestic tourists. The positive trend manifests itself both in domestic and inbound foreign tourism. A 48% increase has been noticed in comparison with 2014. According to the statistics, the number of foreigners has also been systematically growing since 2014, and it has increased by over 50% compared to 2018. Tourism income in 2018 reached 3.1 billion euros. The share of direct tourism economy in GDP is estimated at 2.4%, and, in the case of the broadly understood tourism economy, the share in GDP is approximately 6.1% of GDP. Tourism economy is largely focused on servicing tourist traffic from economically developed European countries. Potential of tourism economy is characterized by the size of the accommodation base, measured by the number of places, which is 206.1 thousand. In 2017, Slovakia recorded the largest number of overnight stays so far, which exceeded the amount of 14 million (Slovenská agentúra).

Table 3 presents the average disposable income of Slovaks, which proves a constant increase and is over 20% higher than the level in Poland and 10.5 percentage points higher in relation to the average in the European Union. Despite the increase in the analyzed figures, the incomes of Slovaks constitute approximately 56.9% of the average income of the inhabitants of the entire European Union.

**Table 3.** The average disposable income of Slovaks in 2014–2019.

|  | 2014 | 2015 | 2016 | 2017 | 2018 | 2019 |
|---|---|---|---|---|---|---|
| GDP per capita in PPS [%] | 53.6 | 55.1 | 55.1 | 55.3 | 56.3 | 56.9 |
| Average disposable income (Euro) | 13.62 | 14.27 | 14.55 | 14.97 | 15.54 | 15.89 |

Source: Eurostat (2020a).

Today, Slovaks travel just like an average EU citizen. However, the majority of their trips take place within the country. On average, Slovaks make three trips a year, one of which is a trip abroad. At the same time, they spend less money on tourism than the inhabitants of Western Europe. Slovaks are believed to spend two times less money abroad than foreign tourists spend in Slovakia. There are still a large number of people in Slovakia who cannot afford to go on a trip. According to Eurostat (2020b), more than 45% of the Slovak population do not have sufficient funds to pay for a longer holiday of at least one week. About 3.4 million Slovaks travel for private purposes. Table 4 presents tourist activity of Slovaks.

**Table 4.** Slovaks' tourist activity in 2014–2018 compared to the activity of the citizens of 27 European Union member states in 2014–2018 (%).

|  | 2014 | 2015 | 2016 | 2017 | 2018 |
|---|---|---|---|---|---|
| EU 27 | 59.2 | 61.0 | 62.3 | 62.4 | 64.4 |
| Slovakia | 53.5 | 60.1 | 66.1 | 70.7 | 73.2 |

Source: Eurostat (2020b).

In 2018, almost $\frac{3}{4}$ of Slovaks went on a tourist trip, i.e., 19.7 percentage points more than in 2014 and 8.8 percentage points more than the average in the European Union countries. Domestic tourism has increased by more than 80%, and the number of tourists coming to Slovakia from abroad has increased from 1.4 million in 2004 to 2.3 million in 2018. It has taken place since Slovakia joined the European Union.

The Slovak tourism policy system presented in Figure 2 indicates the most important entities that are responsible for the implementation of activities for the development of tourism, and therefore including aspects of social tourism policy in Slovakia.

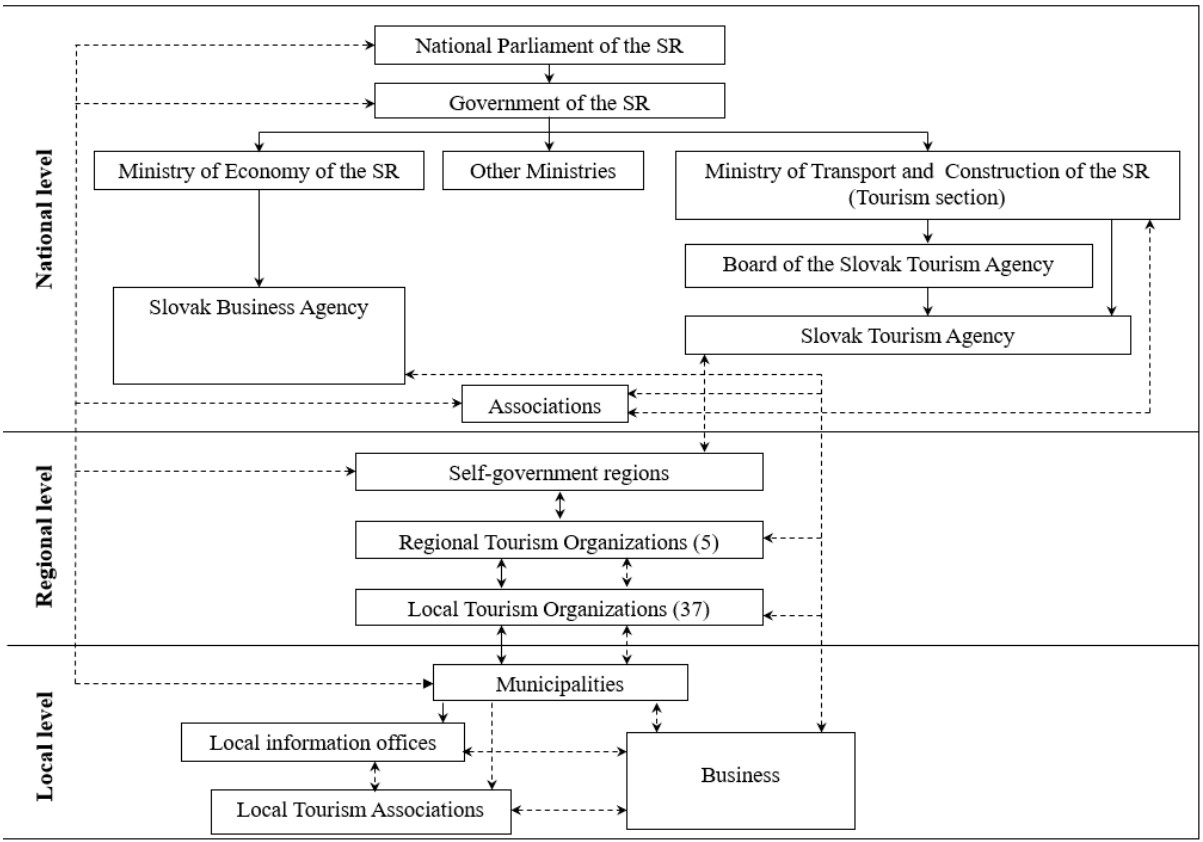

**Figure 2.** Organizations of the Tourism Sector in Slovakia. Source: Elaborated according (Gajdošíková et al. 2016, vol. 230, pp. 405–12).

Until 2011, at the national level, the authority that was responsible for the development of tourism, including the development in the social aspect, was the Ministry of Economy. However, since 2011 it has been the Ministry of Transport, Construction and Regional Development, which results from the fact that the Slovak government has included tourism as a priority area in the system of social and economic policy of the state. The tourism sector of the Ministry is divided into two departments: The Tourism Strategy and Policy Department, and the Tourism Co-operation and Co-ordination Department, with responsibilities including: Creation of conditions for the development of tourism; formulation, implementation, and monitoring of policy; preparation of legislative regulations concerning

tourism; provision of comprehensive statistical data on the development of tourism; and representing the country's interests in international tourism organizations. The Ministry's task is to coordinate and control the activities of the Slovak Tourism Agency, cooperation with research organizations, cooperation with the Council of the Slovak government in the field of coordinated and uniform promotion of Slovakia abroad, foreign cooperation, and fulfillment of agreements resulting from Slovakia's membership in EU and international organizations.

The Slovak Tourist Board (STB) is a government agency funded from the state budget and responsible for promoting and marketing the Slovak Republic as a tourism destination. It markets tourism at the national level, provides information on travel opportunities in the Slovak Republic, promotes a positive image of the country as a tourism destination abroad, supports the sale of tourism products, and implements the EU structural funds in the tourism industry. The STB has eight national branch offices and six foreign offices located in the Czech Republic, Germany, Poland, Austria, the Russian Federation, and Hungary.

At the regional level, tourism responsibilities are devolved to eight self-governing bodies in the regional governments: Brastislava, Trnava, Nitra, Trencin, Zilina, Bankska Bystrica, Presov, and Kosice—and municipalities and towns also play an important role in the development of tourism in their areas. Tourism associations as professional entities also contribute at local and regional levels, primarily in the fields of quality improvement, expanding the range of tourism products and services, professional training, the application of quality standards, and the communication of best practices.

Another public body of social tourism in Slovakia is the Ministry of Labor, Social Affairs and Family, which provides financial support, including subsidies for recreational stays for unemployed Slovak citizens that are recipients of pension benefits. This support is also provided by the Confederation of Trade Unions of the Slovak Republic.

The entities that are indirectly involved in the implementation of tourism policy, mainly the social one in Slovakia, are also national non-governmental organizations and associations dealing with the improvement of the living conditions of Slovaks struggling with difficulties, e.g., the Big Family Club (Klub mnohodetných rodín), the Age Aid Forum (Fórum pre pomoc starším), and the Youth Council of Slovakia (Rada mládeže Slovenska).

Two large hotel chains should also be indicated when mentioning the entities involved in the development of social tourism in Slovakia. The first one is the joint stock company HOREZZA, the main shareholder of which is the state. The company's name is an abbreviation of "Hotel, entertainment and medical facility". The company has a portfolio of 4 4-star hotels/sanatoriums available to every social group. The network offers reduced prices for soldiers and veterans as well as for the disadvantaged (low income). In 2017–2019, veterans and their families got a 25% discount in hotels of the chain. The service included accommodation and breakfast and free access to swimming pools, wellness and spa services, and free treatment (for stays longer than 2 days—for one disease, longer than 5 days—two).

The second of the hotel chains which focuses on social tourism is the largest and oldest hotel chain—SOREA. It is a privately owned entity founded in 1993, with 11 hotels and a total capacity of over 3000 beds. The network packages include holiday stays, e.g., for skiing, spa tourism, and conference tourism. Most of the guests of SOREA hotels are domestic tourists (over 50%). The SOREA network in cooperation with the Ministry of Labor, Social Affairs and Family has created special weekly packages for seniors, for which the government grants EUR 50 per stay (until 2012 it was EUR 70). Every Slovak senior (60+) can apply for such a subsidy once a year only.

A special package for Slovak seniors includes 6 nights with full board included. Another private entity, focused on supporting youth tourist trips, is CKM 2000 Travel. The main mission of this entity since its creation (1999) has been to improve the quality of life of young people, students, and teachers by offering special discounts on railway journeys, coach and air travel, organization of student exchange, work, and study programs, summer

schools, and individual tourism. CKM is a member of EYCA (European Youth Card Association), IYHF (International Youth Hostel Federation), and IATA (International Air Transport Association) (https://www.ckm.sk).

Social tourism in Slovakia became a topic of interest of the country's authorities during the planning period of the tourism policy for 2007–2013. The contribution of tourism to social development became one of the principles of the Slovak tourism policy outlined in the documents "National Tourism Policy" and the "New Strategy for Tourism Development in Slovakia in 2007–2013" and the Act on Support for Tourism (No. 91/2010 Coll.). Then, holiday vouchers, as a tool that enables access to tourism for any social group that experienced poor conditions, were announced to be introduced.

The aim of the tool was to stimulate the demand for domestic tourism, increase the accessibility of tourism to Slovak citizens, especially to socially disadvantaged groups, and to increase utilization rates of Slovak accommodation facilities. However, in spite of the government's efforts, the vouchers were not introduced due to the negative attitude of the Association of Towns and Municipalities of Slovakia (ZMOS), which presented their remark on the tax aspects.

One of the tools that support social tourism in Slovakia aimed at primary and secondary school students are subsidies for the School in Nature and ski courses. Since 1948, in the territory of today's Slovakia, children's stays away from their place of permanent residence were organized for several days. The purpose of these trips was to support the health and increase the physical fitness of children from large cities with polluted air. Initially, they were four-week stays, then they were initially shortened to three weeks, and from 1990 to 5–10 days. Since 2016, these stays are partially subsidized from the state budget, which is guaranteed by law (No. 597/2003 Coll.).

Today, the Ministry of Education, Science, Research, and Sports of the Slovak Republic (MEDU) awards a grant of EUR 100 or EUR 150 (in less developed regions) per student. A 1–4 grade student may receive a grant to support School in Nature once only. Support for the ski course (150 EURO per student) is also provided under the above-mentioned activities. The fund, in turn, is intended to be used by primary school students in grades 5–9 and for secondary school students (Table 5).

**Table 5.** Subsidy from the state budget for primary and secondary school students for tourist purposes (EUR million).

|                    | 2016    | 2017   | 2018    | 2019    |
| ------------------ | ------- | ------ | ------- | ------- |
| Supported activity | 5.0655  | 4.4702 | 5.4909  | 5.6639  |
| Ski course         | 10.7010 | 9.8314 | 12.8525 | 12.8474 |

Source: Derco and Štrba (2020, vol. 23, pp. 2503–6).

A tool that supports social tourism in Slovakia, aimed at retired people, is the state subsidy for co-financing convalescent care stays. According to the provisions of the Act No. 544/2010 on subsidies at the Ministry of Labor, Social Affairs and Family of the Slovak Republic (MLSAF), only a retired person who does not receive income from an employment contract or from business and other activities may be given a partial reimbursement. Most supported stays last a week with full board provided in several objects of accommodation. Quota of the subsidy for a partial reimbursement of the costs of the stay is EUR 50 per person a year.

The latest instrument of tourism policy of Slovakia created to support development of social tourism are holiday vouchers introduced by the government on 1 January 2019. The government of the Slovak Republic announced introduction of the tool in 2010. Holiday vouchers are benefits aimed at supporting domestic tourism in Slovakia. They can only be used within the country for a minimum stay of two nights. The coupon may be given to a person employed at the company for over 2 years, provided that the employer hires more than 49 people. The employer is obliged to reimburse 55% of the employee's holiday

expenses, given that the total amount of the grant may not exceed 275 EUR. A voucher may be given once a year. It can as well be used by spouses and children. The effect of voucher introduction is already noticeable in the statistics of tourist traffic in Slovakia. The number of overnight stays performed by domestic tourists increased by 15.1% per year only, which was the highest increase in the whole European Union.

### 4.3. An Attempt of Comparative Assessment

The presented scope of activities as part of tourism policy in the social aspect in Poland and Slovakia was summarized in Table 6. Common solutions resulting from the activities carried out at the European Union level, in the field of organization and funding of tourist trips, mainly in periods outside of the tourist season for selected social groups, were not included in the table. These activities are aimed mainly at activation of the tourism economy outside of the tourist season.

**Table 6.** The list of basic instruments of internal tourism policy in the field of achieving social goals in Poland and Slovakia.

|  | | Poland | | Slovakia | |
|---|---|---|---|---|---|
|  | Instruments | Entity responsible | Recipients–social groups | Entity responsible | Recipients–social groups |
| **Organizational** | General activities for the development of the tourist market, including tourism demand | Minister responsible for tourism | Society | Minister responsible for tourism | Whole society |
| | Supporting recreation for children and adolescents | Minister responsible for children's physical education and sport | Children and youth | | |
| | Supporting family tourism | Industry organizations | Children and youth, Families | | |
| | Organization of holidays for children and youth | Enterprises and institutions | Children and youth | | |
| | Organization of holidays for employees and their families | Enterprises and institutions | Employees and their families | | |
| | Organization of recreation for all social groups | | | Organization of recreation for every social group | Whole society |
| **Financial** | Financing health tourism | Minister of health | Patients (regardless of their age) | Minister of health | Retired people |
| | Financing the so-called Schools in Nature | Minister of education | Children and youth | Ministry of education | Children and youth |
| | | Local governments | | | |
| | | Social organizations (including scouting inter alia) | | | |
| | Financing holidays for children and youth | Enterprises and institutions | Children and youth | | |
| | Co-financing of sports tourism | | | Ministry of education and sport | Children and youth |
| | Co-financing of holidays for employees and their families | Enterprises and institutions | Employees and their families | | |

**Table 6.** *Cont.*

| | Poland | | | Slovakia |
|---|---|---|---|---|
| Co-financing of holidays for pensioners | | | Ministry of labor and social policy | Pensioners |
| | | | Trade unions | |
| Discounted process for hotel services for selected social groups | Industry organization exploiting accommodation facilities | Children and youth (groups in particular), Families | Tourist enterprises with state capital | Special social groups (soldiers, veterans) |
| Co-financing of stays in hotel facilities for seniors | | | Private enterprise in cooperation with the Minister of labor and social policy | Seniors |
| Co-financing of tourist trips of young people and teachers | | | Ministry of education and sport | Youth (including students) and teachers |
| Co-financing of holidays for poor and/or big families | | | Social organizations | Families |
| Tourist voucher/Holiday voucher | Minister of national tourism Tourist organization | Children and youth under 18 years of age * | Employers hi ring more than 49 people | Employees ** |

\* Once from 1 August 2020 to 31 March 2022. \*\* Once a year (since 2019). Source: Author's own elaboration based on the research.

The organizational and financial instruments related to the social aspects of tourism policy in Poland and Slovakia presented in Table 6 generally relate to the influence of state authorities and cooperating entities on social issues. The economic aspect related to the possibility of obtaining effects by tourism economy entities is of a secondary nature. Both of the above-mentioned countries developed their system of support for tourism consumption, based on the solutions used already in the period of the centrally controlled economy.

The applied instruments relate to similar aspects of support for social groups in accessing tourism. However, they are of a very individual character, appropriate for a given country. It should be noted that in both countries, the ministry responsible for tourism implements a minimum scope of support. In practice, it is limited to influencing economic processes only. Social issues in the field of tourism are implemented by other governmental units: Labor and social policy, health, education, as well as local government units, non-governmental institutions, commercial units, and public employers for the benefit of their employees.

## 5. The Concept of the Social Tourism Policy Model

On grounds of the presented analyzes of tourism policy in terms of the implementation of social goals by the European Union bodies and in two selected countries, an attempt can be made to generalize, consisting in the construction of the concept of a cause-and-effect model of social tourism policy, taking institutional and instrumental elements into account (Figure 3). The model was based on the analysis of solutions applied in two selected European Union countries, which, as indicated before, transferred some of the applied instruments of social tourism policy from the period of centrally planned economies. Despite the lack of a coherent (common, uniform) system of tourism policy within the European Union, solutions regarding the structure of tourism policy entities and the scope of their activities result from the EU legal acts, which were adopted by the analyzed countries into the legal system upon joining the structures of the European Union, and then since 2004, they have implemented new solutions on a regular basis. Therefore, it is

a basic determinant of universality of the presented concept for individual EU countries. Due to the fact that the structure of the presented concept is of a model nature, it is easy to adapt its assumptions to the evaluation of social tourism policy in other European Union countries.

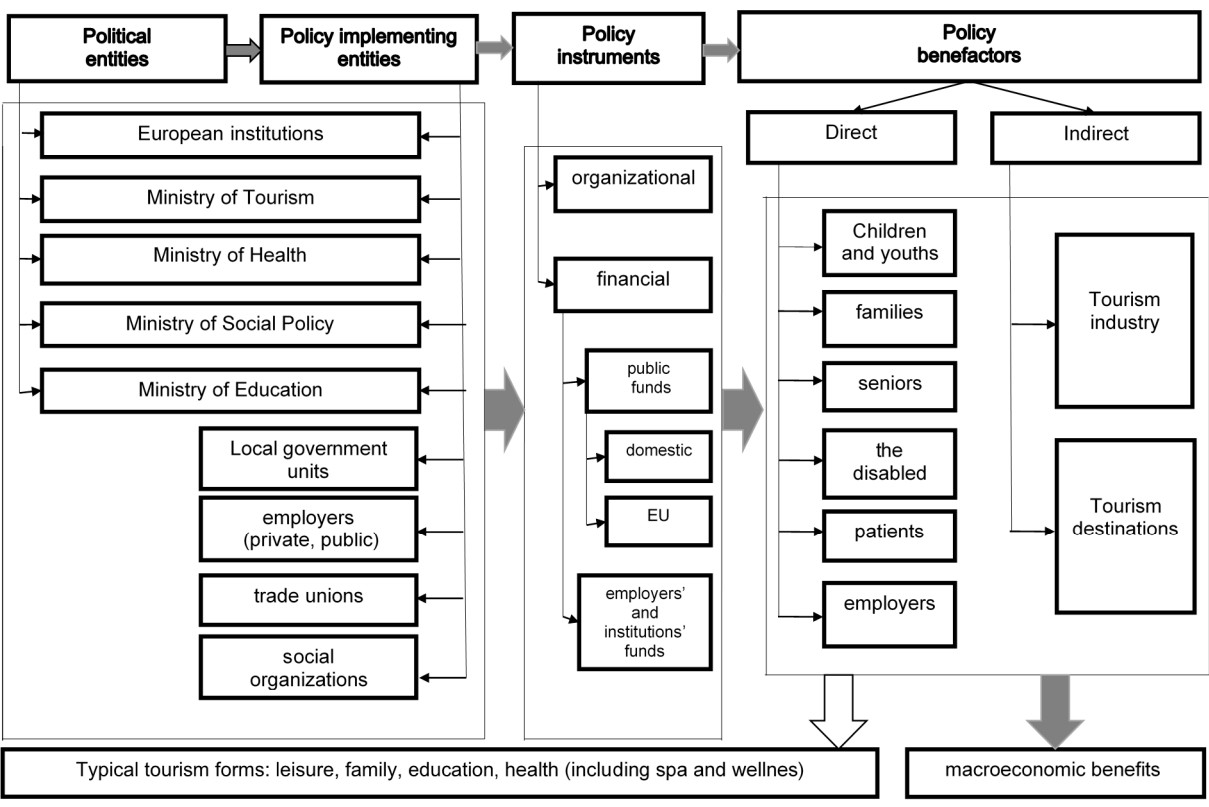

**Figure 3.** Structural model of social tourism policy. Source: Author's own elaboration.

According to the concept presented in Figure 3, European and governmental institutions of a given country are responsible for the tourism policy, the aim of which is to activate the tourism consumption of the society and/or of selected social groups. The tasks of the European Union entities and the national ministry responsible for tourism are mainly to support tourism economy entities through the implementation of demand activating programs. Other public authorities (mainly ministries responsible for social affairs) undertake activities with units that are not entities of politics. Instead, they perform tasks in this field, i.e., with local government units, employers, trade union organizations, and social organizations. Their activity is mainly aimed at supporting certain social groups when accessing tourism. The effects arising in the tourism economy from the point of view of their activities are subservient.

Therefore, social tourism policy is conducted outside of the Ministry of Tourism, whose competences are dominated by strictly economic issues. As part of the task implementation, policy entities, along with the units supporting their activities, take advantage of the instruments mainly of organizational and financial nature. The second group of instruments is related to the possibility of full financing or co-financing of the task from public funds (government and local government) and employing entities.

The whole of the policy is aimed at direct beneficiaries who have the possibility of using organized tourist offers or subsidies for tourism. Beneficiaries are representatives of the classified social groups: Children and youth (including students), families (particularly those with many children and in a difficult financial situation), seniors (particularly retirees), disabled people using the health tourism offer on the basis of referrals from the public

health service or social security institution, employees of enterprises and institutions along with their families.

Indirect effects that result from the conducted social tourism policy are mainly achieved by the entities of direct tourism economy (hotel industry, transport, catering, tourist attractions sector), which create an offer for social groups to which organizational and financial support is provided, as well as other offers, when a tourist voucher is used. Indirect beneficiaries are also tourist destinations to which tourist traffic is directed, and in this case tourists whose stay is financed or co-financed from the funds covered by the programs of its social activation.

The effects of tourism business and tourist destinations are the result of financing typical forms of social tourism, i.e., leisure, family, educational, health, health resorts in particular, possibly also other more detailed forms that are covered by support programs by the European institutions, governments of the Member States or local government units. Thanks to the social tourism policy, not only goals related to influencing direct and indirect beneficiaries of this policy are realized, but also meso- and macroeconomic benefits related to the functioning of the tourism economy in relation to other sectors of the economy and the entire economy, particularly those related to the impact on the labor market, incomes resulting from the tax system (national budget and local government budget).

The presented model is universal. Its structure can be adapted to the detailed institutional and instrumental solutions in individual EU member states. Thus, the assumptions of the model can be transferred to the realities of the tourism economy affected by the COVID-19 crisis (Panasiuk 2020b). Then, the relevant entities of tourism policy responsible for the pandemic mitigation tasks should be interpreted, as well as the applied instruments and the recipients of these actions should be indicated. For instance, the indicated instrument, which is used in Poland, that is a tourist voucher addressed to all children up to 18 years of age, is within the competence of the minister responsible for tourism in cooperation with the national tourist organization, and has a direct impact on tourism demand, mainly in the field of tourism family. However, the aim of this solution is to influence tourism enterprises that will use the travel vouchers, thus increasing the sales of their services and revenues.

*5.1. European Social Tourism Policy*

The presented model concept of social tourism policy has an ideological character and is based on the results of analyses that were carried out in two selected countries. The scope of the policy pursued results from the scope of activities undertaken at the level of the entire European Union and in individual member states, in which the scope of the instruments used varies in type and quantity. In Poland and Slovakia, it is determined by solutions that have remained from the period of the centrally controlled economy and are complemented with some new solutions that are to bring benefits, not only individual ones to the social groups covered by the support, but the ones related to the functioning of the entire tourism economy instead. In the last three financial perspectives of the European Union, in which Poland and Slovakia were already full-fledged members, i.e., in the years (2000)/2004–2006, 2007–2013, and 2014–2020, despite significant funds allocated from the EU budget for tourism purposes, mainly in the perspective (2007–2013), social aspects did not constitute a significant priority (Panasiuk 2014). At this point, it is necessary to settle the question of the existence of a common EU tourism policy, which does not exist in the horizontal dimension. Only selected regulatory actions are taken, the task of which is nothing but the necessary regulation aimed at protecting competition in the tourist services market and consumer protection in the tour operator market.

Another planning period in the EU for 2021–2027 is currently subject to detailed negotiations between the governments of the member states. The previous arrangements prove that tourism goals will not include social aspects. EU financing may cover objectives in the field of protection, development, and promotion of natural heritage and ecotourism (Kot 2018, pp. 10–11). However, it should be recognized that European funds are an

essential and permanent element of tourism policy, as they determine the development of the tourism economy as well as, directly and indirectly, of many other elements of the national economy (Panasiuk 2019).

One of the most important issues that may ultimately determine the place of tourism in EU funding for 2021–2027 is the crisis in the tourism economy caused by the COVID-19 pandemic. The use of structural funds and a specially created fund for the reconstruction of the EU economy should largely support the tourism economy (Panasiuk 2020a), thanks to programs aimed not only at supply, but also at activation of tourism demand, through programs generally aimed at the EU society or selected social groups.

### 5.2. Inference Limitations

Qualitative research methods were used for the undertaken empirical research. On the basis of the conducted analysis of social tourism policy in Poland and Slovakia, the entities of the policy, the scope of the instruments used, and the beneficiaries of the policy were identified. On this basis, an attempt was made to generalize the model of social tourism policy, which has the character of an ideological concept based on the analysis of two sample countries. Due to the research objective of the analysis in these two selected countries, as well as due to the large diversity and scope of the issues addressed, the use of the suggested model requires knowledge on social tourism policy, by incorporating elements that are specific to other EU countries, both institutional and instrumental. The structural concept should make it possible to interpret the scope of the social tourism policy pursued.

The article proves that links with the issues of sustainable development, due to its important role in activities undertaken by the European Union, should be taken into account when shaping the social tourism policy. The issue has not been analyzed in detail. However, it should be pointed out that, especially from the point of view of financing tourism projects from the European Union funds, it is possible to include both social and sustainable tourism development objectives in the tourism policy.

What should be borne in mind is that the research conducted by Paunović and Jovanović (2019) shows that the social aspects of sustainable tourism are of less importance in mountain tourist destinations located in the areas of new EU Member States and EU candidate countries (e.g., Dynřské Mountains—Dinarides) compared to the Alpine destinations (Alps), which are mainly located in the areas of the old EU Member States. This problem has not been studied. However, it may be significant for social aspects in sustainable tourism policy at the level of the European Union and both analyzed member states.

### 5.3. Final Conclusions

As it was mentioned in the introduction, the vast majority of tourism policy instruments in terms of achieving social goals implemented independently in Poland and Slovakia, both by government entities and other entities co-creating this policy, were transferred or modified from the realities of the socialist economy. These are almost all the instruments presented in Table 6. The solutions that were created during the membership of both countries in the European Union were introduced in Slovakia in 2019, and in Poland in 2020 during the pandemic and consist in the activation of domestic tourism. In Slovakia, they include employees of at least medium-sized companies, and in Poland, they include children and young people up to 18 years of age. The aim of these instruments (tourist voucher/coupon) is not only to influence the demand, but most of all to use the service potential of the national tourism economy entities fully. The instruments used at the level of the entire European Union (Calypso, COS-TFLOWS) that are dedicated to the selected social groups of the EU countries, including Poland and Slovakia, are of a similar nature.

The undertaken research goal was achieved by presenting a complete concept of the social tourism policy model in the European Union and its member states. The research questions were also answered by: (1) Identifying and classifying policy entities that implement pro-social activities in the field of promoting access to tourism; (2) selecting entities

from outside of the public sphere that contribute to the social dimension of tourism policy; (3) identification of the instruments used to support the entire society or selected social groups in access to tourism; (4) identifying direct and indirect beneficiaries of social tourism policy. All the analyses indicate that the activities in the field of social tourism policy are conditioned by economic solutions, especially in terms of sources and forms of financing. Moreover, thanks to the implementation of social goals, it is possible to influence economic effects of both entities providing tourist services and macroeconomic effects.

**Author Contributions:** Conceptualization, A.P. and E.W.-S.; responsible for the ideation, A.P.; performed the literature search and data analysis, A.P. and E.W.-S.; drafted and revised the work A.P., E.W.-S.; methodology, A.P., E.W.-S.; validation, A.P., E.W.-S.; formal analysis, A.P., E.W.-S.; investigation, A.P., E.W.-S.; writing—original draft preparation, A.P., E.W.-S.; writing—review and editing, A.P., E.W.-S.; visualization, E.W.-S.; supervision, A.P.; project administration, A.P., E.W.-S., funding acquisition, A.P., E.W.-S. All authors have read and agreed to the published version of the manuscript.

**Funding:** This research received no external funding.

**Institutional Review Board Statement:** Not applicable.

**Informed Consent Statement:** Not applicable.

**Data Availability Statement:** Not applicable.

**Acknowledgments:** The authors would like to thank Jana Kucerová for her inspiration and support in the implementation of this project.

**Conflicts of Interest:** The authors declare no conflict of interest.

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
