# Peer review of "Social Aspects of Tourism Policy in the European Union. The Example of Poland and Slovakia"

_economies, doi:10.3390/economies9010016_

Round 1

Reviewer 1 Report

The socio-economic perspective of the article should be better delineated and rooted in the wider sustainability context (society, economy, environment) as this is the approach promoted by all relevant EU tourism policy documents (for reference, see bibliography below). In this sense, not taking into account the environmental aspects into the study’s approach should be mentioned in the study limitations. Social aspects of sustainable tourism are identified to be of lower importance in mountain DMOs of the Dinarides (consisting of new EU members and EU candidate countries), compared with the DMOs in the Alps (dominated by old EU member states). This points to the clear relevance of social aspects in the sustainable tourism policy on the EU level (Paunović & Jovanović, 2019), and could be of relevance for the case of Poland and Slovakia, as new EU member countries.

Abstract

The first sentence in the abstract should be reformulated. The goal of the policy is not to prove things, but it can be said that the policy is focused on social aspects as means of achieving sustainable development of tourism.

Second sentence-  please be more specific than “it’s (referring to the project) participants”. The meaning is not clear.

Third sentence- “economic development of a state” is a bit imprecise and partial term. What about country or nation? Social tourism policy should be considered as a contribution to the overall and sustainable development, not only economic aspects of development.

The policy and especially tourism policy in the Eu are based on the principles of sustainability (with economic, social and environmental aspects) as well as competitiveness (Estol, Camilleri, & Font, 2018; Paunovic, Dressler, Mamula Nikolic, & Popovic Pantic, 2020)

The question of social tourism policy activities is strongly dependent on governance arrangement which than influence financing modes. This is well documented in the literature (Estol & Font, 2016) , and therefore the direct link between activities and financing is an imprecise interpretation of this issue.

Introduction

What is the share/contribution of tourism in national economies in Slovakia and Poland. This information would be far more relevant than the global average, as the differences between the countries are enormous on the global scale, as well as inside the EU.

Please check if the link should be placed in the footnote or similar, it ruins the text flow.

Please reformulate or delete the sentence “The attitude of the EU bodies to tourism policy has changed over time.” It is too general, cites no references, and the most important defecency is that the public institutions are in charge of creating and implementing policies, so attitudes towards policies are not a relevant research approach.

Methodological assumptions

Please define or reformulate “policy entities”? Maybe it would be better to define them as “public entities”?

Please correct grammatical error: “The conducted research is ended….”, “both economic and social golas”

Please identify the primary data sources used in the articles. Having in mind the wide scope of the article, different types of documents have been used from the two countries. Please name all of the document types used for data collection. The methodological steps should be better explained in the text.

3.2. Tourism policy

Regarding limiting the state’s intervention in the economy, please take into consideration the concept of “governing at arm’s length”, see Bramwell and Lane (2010).

Page 12, line 519: please delete an emply row

Table 6: please check the font in the first row of the table on the left

The main weakness of the article is the insufficiently explained method of organized data collection, processing and analysis, Having in mind the importance of the subject researched and the scope of the results provided, the method should be explained in more detail.

 Good luck with the changes!

Bramwell, B., & Lane, B. (2010). Sustainable tourism and the evolving roles of government planning. In: Taylor & Francis.

Estol, J., Camilleri, M. A., & Font, X. (2018). European Union tourism policy: an institutional theory critical discourse analysis. Tourism Review, 73(3), 421-431. doi:10.1108/tr-11-2017-0167

Estol, J., & Font, X. (2016). European tourism policy: Its evolution and structure. Tourism Management, 230-241.

Paunovic, I., Dressler, M., Mamula Nikolic, T., & Popovic Pantic, S. (2020). Developing a Competitive and Sustainable Destination of the Future: Clusters and Predictors of Successful National-Level Destination Governance across Destination Life-Cycle. Sustainability, 12, 4066. doi:10.3390/su12104066

Paunović, I., & Jovanović, V. (2019). Sustainable mountain tourism in word and deed: A comparative analysis in the macro regions of the Alps and the Dinarides. Acta Geographica Slovenica, 59(2), 59-69.

Author Response

Response to Reviewer 1 Comments

Thank you for reading the text of the article carefully and for all comments. All suggestions were included in the revised and supplemented text. Thank you for your valuable and helpful review. Sincerely, authors

Point 1: The socio-economic perspective of the article should be better delineated and rooted in the wider sustainability context (society, economy, environment) as this is the approach promoted by all relevant EU tourism policy documents (for reference, see bibliography below).

The issues of sustainable tourism development have been included in the text.

The proposed literature sources were used and included in the References

Point 2: In this sense, not taking into account the environmental aspects into the study’s approach should be mentioned in the study limitations. Social aspects of sustainable tourism are identified to be of lower importance in mountain DMOs of the Dinarides (consisting of new EU members and EU candidate countries), compared with the DMOs in the Alps (dominated by old EU member states). This points to the clear relevance of social aspects in the sustainable tourism policy on the EU level (Paunović & Jovanović, 2019), and could be of relevance for the case of Poland and Slovakia, as new EU member countries.

It was included in the content of the article in the Inference limitations (limitations of the study).

Point 3: The first sentence in the abstract should be reformulated. The goal of the policy is not to prove things, but it can be said that the policy is focused on social aspects as means of achieving sustainable development of tourism.

It has been corrected.

Point 4: Second sentence-  please be more specific than “it’s (referring to the project) participants”. The meaning is not clear.

It has been corrected.

Point 5: Third sentence- “economic development of a state” is a bit imprecise and partial term. What about country or nation?

It has been corrected.

Point 6: Social tourism policy should be considered as a contribution to the overall and sustainable development, not only economic aspects of development.The policy and especially tourism policy in the Eu are based on the principles of sustainability (with economic, social and environmental aspects) as well as competitiveness (Estol, Camilleri, & Font, 2018; Paunovic, Dressler, Mamula Nikolic, & Popovic Pantic, 2020)

These issues were included in the text of the article.

Point 7: The question of social tourism policy activities is strongly dependent on governance arrangement which than influence financing modes. This is well documented in the literature (Estol & Font, 2016), and therefore the direct link between activities and financing is an imprecise interpretation of this issue.

These issues has been explained in the article.

Point 8: What is the share/contribution of tourism in national economies in Slovakia and Poland. This information would be far more relevant than the global average, as the differences between the countries are enormous on the global scale, as well as inside the EU.

The information is included in article, however it was also presented in the Introduction.

Point 9: Please check if the link should be placed in the footnote or similar, it ruins the text flow.

It has been changed, the full link is given in the References

Point 10: Please reformulate or delete the sentence “The attitude of the EU bodies to tourism policy has changed over time.” It is too general, cites no references, and the most important defecency is that the public institutions are in charge of creating and implementing policies, so attitudes towards policies are not a relevant research approach.

The sentence has been redrafted.

Point 11: Please define or reformulate “policy entities”? Maybe it would be better to define them as “public entities”?

The sentence has been redrafted.

Point 12: Please correct grammatical error: “The conducted research is ended….”, “both economic and social golas”.

The grammatical error has been corrected.

Point 13: Please correct grammatical error: “The conducted research is ended….”, “both economic and social golas” Please identify the primary data sources used in the articles. Having in mind the wide scope of the article, different types of documents have been used from the two countries. Please name all of the document types used for data collection.

A paragraph on the main sources and their types of collecting information was introduced in the text.

Point 14: The methodological steps should be better explained in the text.

A paragraph concerning the next on stages of the research was added to the text.

Point 15: Regarding limiting the state’s intervention in the economy, please take into consideration the concept of “governing at arm’s length”, see Bramwell and Lane (2010).

The information and footnote were added to the text.

Point 16: Page 12, line 519: please delete an empty row.

It has been corrected.

Point 17: Table 6: please check the font in the first row of the table on the left

It has been corrected.

Point 18: The main weakness of the article is the insufficiently explained method of organized data collection, processing and analysis, Having in mind the importance of the subject researched and the scope of the results provided, the method should be explained in more detail.

The authors accept the Reviewer’s remark. We tired to include these issues in the corrected version of the article.

Reviewer 2 Report

This study analyses the social tourism policies of two countries, Poland and Slovakia, on the basis of which the authors draw up a model.

The following can be identified as strengths:

- The study identifies a knowledge gap: the existence of few studies on social tourism policies.

- The study proposes a model for social tourism policies

The following can be identified as weak points:

- Lack of theoretical support for the methodology used in the design of the model

- Two countries with homogeneous characteristics were used to build the model, namely a common past of planned economies, which compromises its potential for generalization throughout the European Union

- Section 3.3 lacks bibliographic references to support the statements presented

- The impact of the OVID-19 Pandemic on the proposed model should be better explained

Author Response

Response to Reviewer 2 Comments

Thank you for reading the text of the article carefully and for all comments. All suggestions were included in the revised and supplemented text. Thank you for your valuable and helpful review. Sincerely, authors

Point 1: Lack of theoretical support for the methodology used in the design of the model.

Two countries with homogeneous characteristics were used to build the model, namely a common past of planned economies, which compromises its potential for generalization throughout the European Union

The issue has been explained in the article. An attempt was made to objectify the obtained information. Poland and Slovakia have been the EU member states since 2004 and before accession, they adopted the tourism management system which has been in force in the EU countries.

Point 2: Section 3.3 lacks bibliographic references to support the statements presented.

The references to the sources of the literature were added in section 3.3.

Point 3: The impact of the COVID-19 Pandemic on the proposed model should be better explained.

The problem of the COVID-19 pandemic and its impact on the proposed model was explained in the article.

Round 2

Reviewer 1 Report

Thank you for providing the revised version of the manuscript. Below you can find some minor issues which need author's attention:

Line 13: “economy”, not “ecenomy”

Line 14: The two sentences in this line could be merged into one, as they repeat the same thought.

Lines 161-167, 180-192: You might want to consider breaking down this paragraph into more than one sentence. The sentence is too long.

Line 984: “studied”, not “studiem”

Author Response

Thank you very much for pointing out the defects in the text of the article. All suggested changes have been incorporated. The text was subject to linguistic consultation. respectfully, authors